# Dental health assessed using panoramic radiograph and adverse events in chronic kidney disease stage 4–5 patients transitioning to dialysis and transplantation–A prospective cohort study

**Mikko J. Järvisalo**[1,2,3]*, **Viljami Jokihaka**[4], **Markus Hakamäki**[1], **Roosa Lankinen**[1], **Heidi Helin**[4], **Niina S. Koivuviita**[1], **Tapio Hellman**[1], **Kaj Metsärinne**[1]

1 Kidney Centre, Turku University Hospital and University of Turku, Turku, Finland, 2 Department of Anaesthesiology and Intensive Care, Turku University Hospital and University of Turku, Turku, Finland, 3 Perioperative Services, Intensive Care and Pain Medicine, Turku University Hospital and University of Turku, Turku, Finland, 4 Department of Radiology, Dental Radiology, Turku University Hospital, Turku, Finland

* mikko.jarvisalo@tyks.fi

**Data Availability Statement:** The data underlying this study contain potentially identifying participant

## Abstract

### Background and aims

Oral health could potentially be a modifiable risk factor for adverse outcomes in chronic kidney disease (CKD) patients transitioning from predialysis treatment to maintenance dialysis and transplantation. We aimed to study the association between an index of radiographically assessed oral health, Panoramic Tomographic Index (PTI), and cardiovascular and all-cause mortality, major adverse cardiovascular events (MACEs) and episodes of bacteremia and laboratory measurements during a three-year prospective follow-up in CKD stage 4–5 patients not on maintenance dialysis at baseline.

### Methods

Altogether 190 CKD stage 4–5 patients without maintenance dialysis attended panoramic dental radiographs in the beginning of the study. The patients were followed up for three years or until death. MACEs and episodes of bacteremia were recorded during follow-up. Laboratory sampling for C-reactive protein and leukocytes was repeated tri-monthly.

### Results

PTI was not associated with baseline laboratory parameters or C-reactive protein or leukocytes examined as repeated measures through the 3-year follow-up. During follow-up, 22 patients had at least one episode of bacteremia, but only 2 of the bacteremias were considered to be of oral origin. PTI was not associated with incident bacteremia during follow-up. Thirty-six patients died during follow-up including 17 patients due to cardiovascular causes.

information and cannot be shared publicly. Future data access requests should be sent to the Ethics Committee of Southwest Finland Hospital District (eettinen.toimikunta@tyks.fi) or the Department of Nephrology and the Informatics Department of Turku University Hospital via the corresponding author.

**Funding:** This work was supported by research grants from Finska Läkaresällskapet (KM), https://fls.fi/sallskapet/ and Perklén Foundation (KM), http://www.foundationweb.net/perklen/, Helsinki, Finland. The grants were used to collect and analyze the data. The funding bodies were not involved with the design of the study or interpretation of the data.

**Competing interests:** The authors have declared that no competing interests exist.

During follow-up 42 patients were observed with a MACE. PTI was independently associated with all-cause (HR 1.074 95% CI 1.029–1.122, p = 0.001) and cardiovascular (HR 1.105, 95% CI 1.057–1.157, p<0.0001) mortality, as well as, incident MACEs (HR 1.071 95% CI 1.031–1.113, p = 0.0004) in the multivariable Cox models adjusted for age and kidney transplantation or CKD treatment modality during follow-up.

## Conclusions

Radiographically assessed dental health is independently associated with all-cause and cardiovascular mortality and MACEs but not with the incidence of bacteremia in CKD stage 4–5 patients transitioning to maintenance dialysis and renal transplantation during follow-up.

## Introduction

Patients with advanced chronic kidney disease (CKD) have a high prevalence of oral infections and impaired overall oral health compared to the general population [1, 2]. Former studies in CKD patients, including stage 4–5 patients with or without maintenance dialysis and kidney transplant recipients have shown that the remaining number of teeth is inversely associated with the risk for subsequent adverse coronary events and death [3]. The most important cause of tooth loss in middle-aged and older individuals, periodontitis [4], is associated with sustained low-grade infection and inflammation which have been suggested to connect oral health impairment to accelerated atherogenesis, diabetes, stroke and coronary artery disease in affected individuals including those on maintenance dialysis [5–8] Periodontitis has also been shown to be associated with mortality in CKD stage 3–5 patients and kidney transplant recipients [9, 10].

Previous studies have shown that measures of oral health, including radiographic indices are associated with coronary artery disease (CAD), cardiovascular events and sudden cardiac death in patients without pronounced CKD [5, 11–13]. Oral health is a part of the overall assessment in evaluating patient suitability for kidney transplantation and panoramic radiographs (DPR) are taken in every patient at our Kidney Centre, prior to dialysis treatment and/ or assigning them to the kidney transplant waiting list. Oral health screening is performed to prevent bacteremia of oral origin in this highly comorbid patient group some of whom receive transplants and require immunosuppressive medication further increasing susceptibility for severe infections. Previous studies have shown that stage 4–5 CKD patients have a significantly higher risk for hospitalization due to infection of any source including bloodstream infections and carry significantly higher risk for infection related mortality compared to subjects with more preserved kidney function [14].

Despite improvements in medical care, the risk of mortality remains high in patients with advanced CKD and novel strategies for improving the prognosis are needed. Oral health could potentially be a modifiable risk factor for adverse outcomes in CKD patients transitioning from conservative treatment to maintenance dialysis and transplantation. Therefore, we aimed to study the association between an index of radiographically assessed oral health, Panoramic Tomographic Index (PTI), and cardiovascular and all-cause mortality, major adverse cardiovascular events (MACEs) and episodes of bacteremia and laboratory measurements during a three-year prospective follow-up in CKD stage 4–5 patients not on maintenance dialysis at baseline.

## Methods

### Study protocol

210 consecutive patients referred to the predialysis outpatient clinic of Turku University Hospital were recruited between August 2013 and September 2017 to the Chronic Arterial Disease, quality of life and mortality in chronic KIDney injury (CADKID)–study. The study population target for the main CADKID study was set to a minimum of 200 patients in the beginning of the recruitment. CADKID is an ongoing, prospective, follow-up study assessing arterial disease, quality of life, mortality and their predictors in patients with chronic kidney disease (CKD-KDIGO 4–5) (http://www.ClinicalTrials.gov NCT04223726). Inclusion criteria for the CADKID study participants were ≥ 18 years of age and CKD stage 4–5 with an estimated glomerular filtration rate (eGFR) < 30 ml/min per 1.73 m2 calculated using the Chronic Kidney Disease Epidemiology Collaboration (CKD-EPI) equation.

The present study is a pre-specified report from the CADKID study. No power calculations were performed for the present sub-study as the main CADKID study examines several primary and secondary endpoints and their risk factor associations. As the focus of the present study was the association between PTI and adverse outcomes, 20 patients with unavailable DPR data for PTI calculation were excluded. The final study cohort comprised 190 patients.

CKD stage 4–5 patients are followed up regularly at the Kidney Centre predialysis outpatient clinic at 1–3 month intervals. All patients receive verbal oral health education/are interviewed on potential dental problems at the predialysis outpatient clinic and are recommended to visit a dentist yearly. No regular follow-up by a dentist was included in the study protocol and DPR was only systematically assessed at study baseline. No data was available on the visits of the patients to private sector dentists. The individual patient data was collected from the hospital's patient documents and during the study and clinical control visits. The data from the hospital software were combined and the patient identity numbers removed before the statistical analyses. The patients were followed up for three years or until death. Major adverse cardiovascular events (MACE) defined as a composite of cardiovascular death, myocardial infarction, stroke and coronary artery revascularization were recorded. To collect the episodes of bacteremia, all symptomatic febrile episodes with positive blood culture data were recorded during follow-up.

### Dental radiography

DPR was scheduled for 191 patients in the beginning of the CADKID study by the treating nephrologists. One of the patients for whom DPR was scheduled did not attend the scan due to patient related issues. In 19 patients of the main CADKID study cohort (n = 210) DPR was not scheduled due to the patients having been examined by a certified dentist (mostly including DPR) in the private dental care sector within 6 months prior to study recruitment. These data were, however, not available for inclusion in the present study. Therefore, 190 patients were included in the analyses for DPR.

The panoramic radiographs were analyzed by a dentist having experience in interpreting DPRs and undergoing oral and maxillofacial radiology specialization training in the research hospital. As described by Karhunen et al. the number of caries lesions, periapical lesions, pericoronitis, cystic lesions (> 4mm), furcation lesions (III grade), vertical bony pockets (> 3mm), and residual roots were summed in each individual radiograph to constitute the PTI [12]. Intra- and inter-observer variabilities of PTI measurements were assessed by reanalyzing 40 randomly selected DPR images blinded to the results of the first analysis and

calculating the intraclass correlation coefficients (ICC). The intra-observer ICC was 0.92 and the inter-observer ICC 0.80 (mean difference between measurements 1 and 2, respectively).

## Statistical methods

Results are presented as mean ± standard deviation (SD) for the normally distributed variables and as median [inter-quartile range (IQR)] for skewed variables. Normality in continuous covariates was tested with the Kolmogorov-Smirnov and Shapiro-Wilk tests. Univariate correlations between the laboratory variables at baseline and PTI were examined by calculating Spearman's correlation coefficients. The association between PTI as a dichotomous variable (PTI = 0 vs. PTI ≥ 1) and C-reactive protein (CRP) and leukocyte values measured every 3 months during the 3-year follow-up were studied using repeated measures analysis of variance.

Univariate associations between DPR findings and outcomes [all-cause mortality, cardiovascular mortality, MACE and bacteremia] were studied using Cox proportional hazards models. Demographic variables known to be associated with mortality in the CADKID study cohort, namely age, diabetes and coronary artery disease [15] were included in the multivariable Cox proportional hazards models in addition to PTI. As the number of mortality events during follow-up was 36 and number of MACEs was 42, the number of covariates included in the first multivariable Cox model was limited to four to avoid overfitting. A second, respective, multivariable Cox model was further adjusted for the treatment modality for CKD at the end of study or death (conservative, peritoneal dialysis, hemodialysis, kidney transplantation) in addition to age, diabetes, CAD and PTI. Due to the limited number of outcomes during follow-up the number of covariates included in the multivariable models had to be restricted to the fore mentioned variables to avoid overfitting. Furthermore, the association between PTI and outcomes was examined in models adjusted with dentist referral following the DPR and age. The intra-observer and inter-observer repeatability of the PTI measurements was examined by calculating the ICCs using Proc Mixed (SAS Institute Inc.). The ICC, calculated as, ICC = Common Covariance Estimate / (Residual Variance Estimate + Common Covariance Estimate), is an estimate of the reliability of the measurement and varies from 0 (no reliability) to 1 (total reliability, when test = retest measure). All statistical analyses were performed using statistical analysis system, SAS version 9.3 (SAS Institute Inc., Cary NC). All tests were two-sided and $p < 0.05$ was considered statistically significant.

## Statement of ethics

The study protocol (reference number T05/024/20) was approved by the Turku University Clinical Research Center and Turku University Hospital scientific ethics review board and the Hospital district of Southwest Finland. All procedures were in accordance with the Helsinki Declaration. All patients gave written informed consent before entering the study.

## Results

The study included 190 patients with DPR imaging performed at study baseline. The clinical characteristics and laboratory measurements of the study patients in the beginning of the study are shown in Table 1 and the DPR findings in Table 2. Median age of the study patients was 65 (52–73) years and 65 of the patients (34.2%) were women. A total of 83 patients (43.7%) had been diagnosed with diabetes and 25 patients (13.2%) with CAD prior to study recruitment. Only 12.1% of the patients were smokers. Median eGFR was 12 (11–15) ml/min per 1.73 m$^2$, median HbA1c was 5.4 (5.1–6.5) %, median cardiac Troponin T was 33 (21–62) ng/l and median CRP was low 2.0 (1.0–6.0) mg/l. (Table 1) Of the DPR findings periapical

**Table 1. Demographic and laboratory characteristics.** Values are presented as mean ± SD for normally distributed variables and median (IQR) for skewed variables.

| | |
|---|---|
| Number of subjects | 190 |
| Female subjects (n/%) | 65/34.2% |
| Male subjects (n/%) | 125/65.8% |
| Age (years) | 65 (52–73) |
| Diabetes (n/%) | 83 (43.7%) |
| Coronary artery disease (n/%) | 25 (13.2%) |
| Antihypertensive medication (n/%) | 187 (98.4%) |
| Smokers (n/%) | 23 (12.1%) |
| Body mass index (kg/m$^2$) | 27.8 (24.5–30.9) |
| Systolic blood pressure (mmHg) | 151 (137–166) |
| Diastolic blood pressure (mmHg) | 81±14 |
| Creatinine (μmol/l) | 412±103 |
| eGFR (ml/min) | 12 (11–15) |
| Urea (mmol/l) | 23.6±6.4 |
| Hemoglobin (g/l) | 114±12 |
| Leukocytes (E$^9$/l) | 6.9±2.0 |
| C-reactive protein (mg/l) | 2.0 (1.0–6.0) |
| Erythrocyte sedimentation rate (mm/h) | 31 (18–46) |
| Albumin (g/l) | 34.8 (32.1–36.9) |
| Sodium (mmol/l) | 141 (140–143) |
| Potassium (mmol/l) | 4.3±0.5 |
| HbA1c (%) | 5.4 (5.1–6.5) |
| Troponin T (ng/l) *n = 182* | 33 (21–62) |

eGFR = Estimated glomerular filtration rate.

lesions were most common (35.8%), followed by carious lesions (24.7%), whereas no patient had >4mm cysts. The prevalence of DPR findings is shown in Fig 1. Median PTI was 1 (0–2). Of the total 109 residual roots detected in the current study 50% (55) were interpreted as having concomitant periapical periodontitis. Due to the DPR findings, 27 patients were referred to a dentist for full dental assessment, but one patient declined.

PTI was not associated with baseline laboratory parameters of interest such as eGFR (r = 0.08, p = 0.29), baseline CRP (r = -010, p = 0.17), leukocytes (r = 0.05, p = 0.57), erythrocyte sedimentation rate (r = 0.08, p = 0.26) or cardiac troponin T (r = 0.06, p = 0.45). Furthermore, PTI was not correlated with age (r = 0.03, p = 0.71) or BMI (r = -0.08, p = 0.30) at baseline. CRP and leukocytes examined as a repeated measures (samples taken every 3 months

**Table 2. Panoramic radiograph findings.**

| | |
|---|---|
| Carious lesions | 24.7%, (range 0–5) |
| Residual roots | 10.5%, (range 0–20) |
| Periapical lesions | 35.8% (range 0–20) |
| Furcation lesions | 5.8% (range 0–5) |
| Pericoronitis | 7.4% (range 0–2) |
| Vertical bony pockets (>3mm) | 5.8% (range 0–4) |
| Cysts (>4mm) | 0% |
| Panoramic tomographic index ≥1 | 57.4% (range 0–33) |

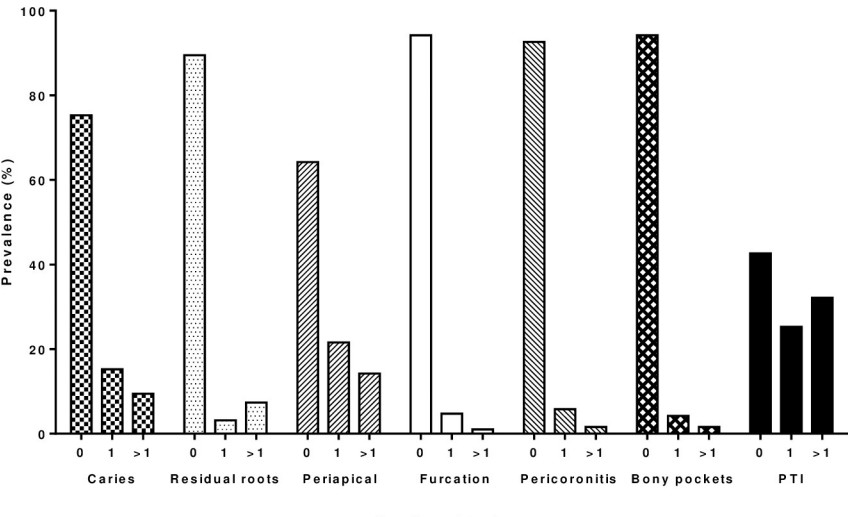

**Fig 1. Prevalence of panoramic radiograph findings at baseline.** PTI = Panoramic tomographic index.

during the 3-year follow-up) were not associated with PTI ≥1 (CRP: β = 1.63, p = 0.26; leukocytes: β = 0.14, p = 0.60). There were no significant differences between patients with or without diabetes in any of the DPR measures at baseline [diabetes: PTI 1 (0–3) vs. others 1 (0–2), p = 0.20] or patients on different treatment modalities for CKD at the end of follow-up [conservative treatment: 1 (0–3), peritoneal dialysis: 1 (0–2), hemodialysis: 1 (0–2), transplantation: 1 (0–2), p = 0.49]. Patients with CAD had a higher prevalence of >3mm vertical bony pockets compared to those without CAD (4 patients (16%) vs. 7 patients (4%), p = 0.02) but otherwise radiographic measures did not differ between the subgroups. Smoking patients tended to have a higher prevalence of residual roots (5 patients (22%) vs. 15 patients (9%), p = 0.07) and a higher PTI compared to non-smoking patients [2 (0–6) vs. 1 (0–2), p = 0.01]. Patients that were referred to the dentist following DPR had significantly higher PTI compared to others [1 (1–3) vs. 1 (0–2), p = 0.003].

During the 3-year follow-up, 22 patients had at least one episode of bacteremia (range 0–5), but only two of the bacteremias were considered to be of oral origin. The blood culture findings are shown in S1 Table. The source of the first bacteremia was skin/soft tissues in 10 (45%), urinary tract in 3 (14%), abdominal in 5 (23%), dental in 2 (9%), catheter-related bloodstream infection in 1 (5%) and respiratory in 1 (5%) patient(s), respectively. The median time to the first bacteremia was 796 (IQR 432–883, range 31–1056) days. None of the DPR measures including PTI (HR 0.862) 95%CI 0.721–1.030, p = 0.10) were associated with incident bacteremia during follow-up in univariate Cox models. Five of the patients in the current study were, on continuous prophylactic antibiotics at baseline (Trimethoprim, for recurrent urinary tract infection, n = 1; sulfatrimethoprim, for Pneumocystic jirovecii prophylaxis, n = 1; phenoxymethylpenicillin n = 1, for recurrent cellulitis; ciprofloxacin, for recurrent urinary tract infection, n = 1 and Clindamycin, for recurrent cellulitis, n = 1). One of the patients with continuous antibiotic prophylaxis had an incident bacteremia during follow-up.

During follow-up 150 patients started dialysis, and 48 received a kidney transplant. 36 patients died during follow-up including 17 patients (47% of all deaths) due to cardiovascular causes. The other causes of death were infection (7/19%), malignancy (6/17%), trauma (4/11%), gastrointestinal (1/3%) and urinary (1/3%). During follow-up 42 (22%) patients were observed with a MACE. Of the surviving patients, 28 were on conservative treatment, 35 on

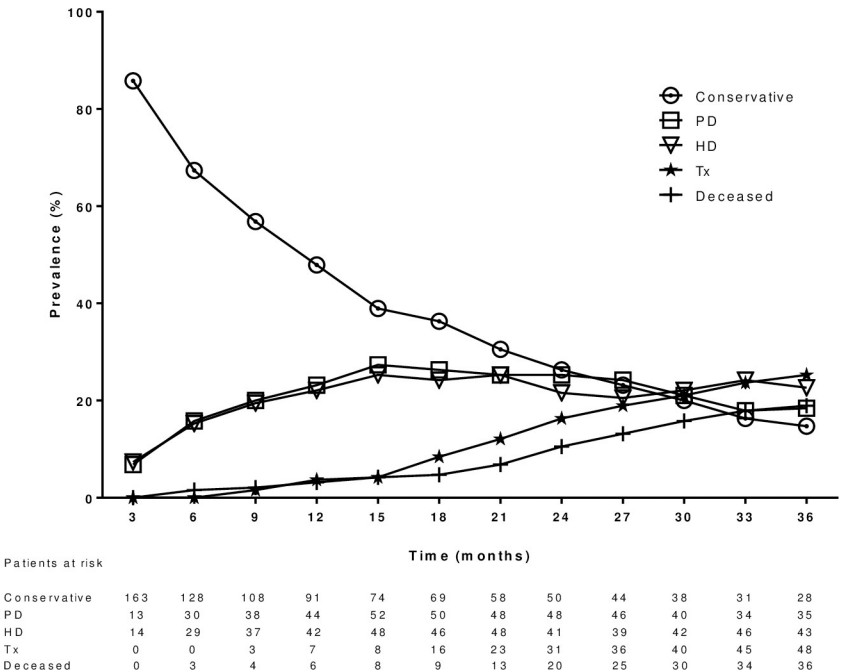

**Fig 2. The clinical course of the study patients through follow-up.** PD = Peritoneal dialysis; HD = hemodialysis; Tx = Kidney transplantation.

peritoneal dialysis, 43 on hemodialysis and 48 had a functional kidney transplant at the end of follow-up (Fig 2). None of the kidney transplants were lost due to rejection during follow-up.

PTI was associated with all-cause (HR 1.070, 95% CI 1.025–1.117, p = 0.002) and cardiovascular (HR 1.104, 95% CI 1.056–1.154, p<0.0001) mortality and MACE (HR 1.071, 95% CI 1.032–1.112, p = 0.0003) in the univariate Cox proportional hazards models. Furthermore, the association between PTI and all-cause (HR 1.072 95% CI 1.026–1.120, p = 0.002) and cardiovascular (HR 1.102, 95% CI 1.050–1.155, p<0.0001) mortality, as well as, MACE (HR 1.062 95% CI 1.021–1.104, p = 0.003) remained significant in the multivariable Cox models adjusted for age, diabetes and coronary artery disease. The results were essentially the same when treatment modality at death or at the end of follow-up was included in the model (Table 3). The association between PTI and outcomes (mortality: HR 1.152 95% CI 1.026–1.139, p = 0.004; MACE: HR 1.066 95% CI 1.020–1.115, p = 0.005) also remained significant when the models were adjusted with age and dentist referral following the DPR.

## Discussion

Our current findings show, for the first time that radiographically assessed oral health in the predialysis phase of advanced kidney disease is independently associated with mortality and incident MACEs during a follow-up up to three years in CKD patients transitioning to maintenance dialysis and/or kidney transplantation.

While extensive data is available on the association of oral health and adverse outcomes in the general population, data limited to CKD patients remains scarce and controversial. In a previous study on CKD patients including kidney transplant recipients and patients on maintenance dialysis the number of teeth was associated with all-cause mortality, whereas, Total dental index [11], a measure of inflammatory burden caused by caries, periodontitis, periapical lesions, and pericoronitis of partially erupted or impacted teeth or Periodontal inflammatory

**Table 3. Univariate and multivariable models.**

| Variable | All-cause mortality | | | Major adverse cardiovascular event | | |
|---|---|---|---|---|---|---|
| | Hazard ratio | 95%CI | p | Hazard ratio | 95%CI | p |
| **Univariate models** | | | | | | |
| PTI | 1.070 | 1.025–1.117 | **0.002** | 1.071 | 1.032–1.112 | **0.0003** |
| Diabetes | 2.151 | 1.100–4.205 | **0.03** | 2.880 | 1.516–5.473 | **0.001** |
| Coronary artery disease | 2.540 | 1.194–5.403 | **0.02** | 2.746 | 1.380–5.465 | **0.004** |
| Age | 1.075 | 1.040–1.112 | **<0.0001** | 1.038 | 1.011–1.064 | **0.005** |
| Treatment modality | | | | | | |
| Conservative | Reference | | | | | |
| Peritoneal dialysis | 0.945 | 0.401–2.225 | 0.90 | 1.593 | 0.627–4.047 | 0.33 |
| Hemodialysis | 1.054 | 0.473–2.345 | 0.90 | 2.008 | 0.844–4.776 | 0.12 |
| Tansplantation | N/A* | N/A | N/A | 0.451 | 0.132–1.541 | 0.20 |
| **Multivariable models** | | | | | | |
| **Covariates included in the model: PTI, age, diabetes, coronary artery disease** | | | | | | |
| PTI | 1.072 | 1.026–1.120 | **0.002** | 1.062 | 1.021–1.104 | **0.003** |
| Age | 1.082 | 1.043–1.122 | **<0.0001** | 1.038 | 1.011–1.066 | **0.006** |
| Diabetes | 2.107 | 1.043–4.256 | **0.04** | 2.504 | 1.286–4.875 | **0.007** |
| Coronary artery disease | 1.178 | 0.536–2.590 | 0.68 | 1.628 | 0.794–3.341 | 0.18 |
| **Covariates included in the model: PTI, age, diabetes, coronary artery disease and treatment modality for chronic kidney disease in the end of the study** | | | | | | |
| PTI | 1.072 | 1.022–1.123 | **0.004** | 1.068 | 1.024–1.114 | **0.002** |
| Age | 1.070 | 1.030–1.112 | **0.0006** | 1.039 | 1.009–1.070 | **0.01** |
| Diabetes | 1.722 | 0.850–3.490 | 0.13 | 2.261 | 1.158–4.415 | **0.02** |
| Coronary artery disease | 1.093 | 0.497–2.401 | **0.049** | 1.434 | 0.696–2.958 | 0.33 |
| Treatment modality | | | | | | |
| Conservative | Reference | | | | | |
| Peritoneal dialysis | 1.447 | 0.594–3.524 | 0.42 | 1.970 | 0.756–5.133 | 0.17 |
| Hemodialysis | 1.688 | 0.725–3.930 | 0.22 | 2.648 | 1.076–6.516 | **0.03** |
| Transplantation | N/A* | N/A | N/A | 1.170 | 0.300–4.558 | 0.82 |

*, No deaths in the transplantation group during follow-up.

burden index were not [3]. Advanced periodontitis has previously been shown to be associated with mortality in CKD stage 3–5 patients [10] and also in kidney transplant recipients with a functioning transplant during a follow-up of 15 years [9].

It seems that oral health, measured radiographically in the present study, is associated with clinical outcomes in patients with advanced CKD. The risk of mortality remains high in patients with advanced CKD and novel strategies for improving the prognosis are needed as several risk factor interventions have failed to improve clinical outcomes in this highly comorbid patient group. Patients with CKD have a 10- to 30-fold increased risk for cardiovascular mortality compared to age-matched populations [16]. Traditional cardiovascular risk factors such as diabetes, smoking, hyperlipidemia and hypertension explain only part of the increased risk in CKD patients [17]. Oral health could therefore potentially be a modifiable risk factor for adverse outcomes in CKD.

The association between PTI and outcomes remained significant after adjusting for dentist referral following DPR imaging. This may be explained by the fact that the patients who have significant PTI findings might be the ones who neglect dental self-care even following the dentist appointment and associated patient education. This would be in line with previous data showing that despite prior oral health education almost half of renal transplant recipients have

residual dental caries after transplantation [18]. In one previous study on CKD patients, oral health was reported to improve during follow-up compared to the predialysis stage, but mortality (45%) and loss to follow-up was high leading to potential bias as baseline comparisons between patients completing and not completing the study were not performed [19]. However, it would be tempting to speculate that the improvement in oral health had been a protective factor in the survived patients of the study by Nylund et al. Therefore, enhanced dental health patient education by nephrologists and oral professionals might be beneficial to increase the quality of dental self-management and even survival in patients with advanced CKD. Randomized data on the benefits of treating periodontal inflammation in CKD patients to reduce systemic inflammation and cardiovascular risk is currently lacking.

Prior available data on the association between oral health and graft survival after kidney transplantation is scarce and inconsistent [19, 20]. It has been argued that poor oral health and self-care observed in a large proportion of CKD 5 patients, with a quarter of patients reporting never brushing their teeth and only 11% ever using dental floss [21], may not be a risk factor for allograft rejection but rather an indicator of concomitant weak adherence to post-transplantation treatment [22]. In the present study all allografts remained functional until study end and we were therefore unable to examine the association between kidney transplant prognosis and PTI.

In the present study there were no differences in PTI between patients with or without diabetes, which, is in controversy with some previous studies [2]. The reason for this finding remains unclear. It is, however, possible that dental health is nowadays managed more properly in diabetic patients who have regular outpatient controls as qlycemic control is known to be associated with dental health [23].

Previous studies have suggested that systemic low-grade inflammation is the link between poor oral health and systemic diseases, including cardiovascular sequelae [24, 25]. PTI or any other of its constituent measures, however, were not associated with baseline laboratory variables including CRP, leukocytes or erythrocyte sedimentation rate, all of which could be thought to be elevated in states of chronic infection or inflammation. Furthermore, CRP and leukocytes examined as repeated measures through follow-up were not associated with PTI and none of the dental measures were associated with incident bacteremia during follow-up. Only a minor proportion of the bacteremias observed were of oral origin, which probably explains the lack of an association between DPR findings and incident bacteremia. The low-grade inflammation in periodontal infections, however, has been reported to manifest also with low CRP ($< 3$ mg/L) [25]. Considering that low-grade inflammation is also a common feature of CKD [26], the sensitivity of the employed laboratory measures might not have been sufficient to differentiate the inflammation caused by oral infectious load from the influence of other comorbidities in the current study population. The use of prophylactic antibiotics may reduce the incidence of bacteremia. Only five of the patients of in the current study were, on continuous prophylactic antibiotics and one of these patients had an incident bacteremia during follow-up. Unfortunately, no data on occasional antibiotics administered such as one-time perioperative prophylaxis at operations or procedures or other means of individual dental prophylaxis, which are known affect the risk of incident bacteremia were available.

The main limitation of our study was that DPR imaging was only performed in the beginning of the study and is therefore a cross-sectional measure, in spite of the prospective patient follow-up. Furthermore, the study protocol did not include a clinical dental assessment by a dentist at baseline or during follow-up. These limitations in the study protocol are a potential source of bias in terms of the association between dental health and incident adverse events. All patients receive verbal oral health education/are interviewed on potential dental problems at the predialysis outpatient clinic and are recommended to visit a dentist yearly. At the time

of the study most patients were instructed to visit a private sector dentist and only those with obvious dental symptoms/imaging findings were referred to a dentist at the research hospital. No data was available on the visits of the patients to private sector dentists. The number of patients included in the analysis is somewhat limited, but the CADKID study recruited all consecutive willing CKD stage 4–5 patients referred to the Turku University Hospital Kidney Centre between 2013 and 2017 and 90% of the CADKID study patients had DPR data. The patients were followed-up prospectively and regularly at our Kidney Center, and the data are of high quality and comprehensive. Due to the limited number of adverse events during follow-up variables included in the multivariable models had to be limited to avoid overfitting which may increase the risk for unrecognized confounders. As power calculations were not performed, it is possible that the study was underpowered for some of the analyses performed. The single center observational study design does not allow for determination of causality and increases the potential for selection bias. The findings, however, were distinct for all outcome measures and limited sample size is not likely to detract the validity of our main findings concerning the association between PTI and adverse outcomes. Our results, therefore, show that PTI offers an inexpensive means to noninvasively screen for impaired dental health in patients with advanced CKD. However, it should be noted that although radiographic assessment seemed to detect poor oral health in patients with advanced CKD in the present study, PTI among other radiographic measures are not effective in evaluating every aspect of oral health, for instance acute periodontal inflammation. The dental examination incorporating both radiographic and clinical assessment would thus be more sensitive in evaluating the oral infectious status.

In conclusion, the current study shows for the first time that radiographically assessed dental health is independently associated with all-cause and cardiovascular mortality and MACEs but not with the incidence of bacteremia in CKD stage 4–5 patients transitioning to maintenance dialysis and renal transplantation during follow-up. Our current data suggest that DPR imaging offers a noninvasive and inexpensive tool for dental health associated mortality and cardiac risk stratification in patients with advanced CKD and should be included in the routine clinical assessment of CKD stage 4–5 patients.

## Supporting information

**S1 Table. Blood culture findings.**
(DOCX)

## Acknowledgments

The authors are grateful to Mrs Eliisa Löyttyniemi, MSc, for statistical consultation.

## Author Contributions

**Conceptualization:** Mikko J. Järvisalo, Viljami Jokihaka, Markus Hakamäki, Roosa Lankinen, Heidi Helin, Niina S. Koivuviita, Tapio Hellman, Kaj Metsärinne.

**Data curation:** Mikko J. Järvisalo, Viljami Jokihaka, Markus Hakamäki, Roosa Lankinen, Heidi Helin, Niina S. Koivuviita, Tapio Hellman, Kaj Metsärinne.

**Formal analysis:** Mikko J. Järvisalo, Tapio Hellman.

**Funding acquisition:** Kaj Metsärinne.

**Investigation:** Mikko J. Järvisalo, Viljami Jokihaka, Markus Hakamäki, Roosa Lankinen, Heidi Helin, Niina S. Koivuviita, Tapio Hellman, Kaj Metsärinne.

**Methodology:** Mikko J. Järvisalo, Viljami Jokihaka, Markus Hakamäki, Roosa Lankinen, Heidi Helin, Niina S. Koivuviita, Tapio Hellman, Kaj Metsärinne.

**Project administration:** Mikko J. Järvisalo, Kaj Metsärinne.

**Visualization:** Tapio Hellman.

**Writing – original draft:** Mikko J. Järvisalo.

**Writing – review & editing:** Viljami Jokihaka, Markus Hakamäki, Roosa Lankinen, Heidi Helin, Niina S. Koivuviita, Tapio Hellman, Kaj Metsärinne.

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
