## [Decision Letter · Decision Letter 0]

3 Aug 2021

PONE-D-21-11023

Dental Health Assessed Using Panoramic Radiograph and Adverse Events in Chronic Kidney Disease Stage 4-5 Patients Transitioning to Dialysis and Transplantation –A Prospective Cohort Study.

PLOS ONE

Dear Dr. Järvisalo,

Thank you for submitting your manuscript to PLOS ONE. After careful consideration, we feel that it has merit but does not fully meet PLOS ONE’s publication criteria as it currently stands. Therefore, we invite you to submit a revised version of the manuscript that addresses the points raised during the review process.

We look forward to receiving your revised manuscript.

Kind regards,

Denis Bourgeois

Academic Editor

PLOS ONE

Journal Requirements:

**3. **Please ensure that you refer to Figure 1 in your text as, if accepted, production will need this reference to link the reader to the figure.

Reviewers' comments:

Reviewer's Responses to Questions

**Comments to the Author**

1. Is the manuscript technically sound, and do the data support the conclusions?

Reviewer #1: Partly

Reviewer #2: Yes

2. Has the statistical analysis been performed appropriately and rigorously? 

Reviewer #1: No

Reviewer #2: Yes

3. Have the authors made all data underlying the findings in their manuscript fully available?

Reviewer #1: No

Reviewer #2: No

4. Is the manuscript presented in an intelligible fashion and written in standard English?

Reviewer #1: Yes

Reviewer #2: Yes

5. Review Comments to the Author

Reviewer #1: Thank you for letting review this interesting article.

Introduction :

- It would be wise to include more recent bibliographic references.

- Only periodontal diseases are discussed, whereas bacteremia origins are not limited to periodontal diseases.

Methods :

- You state that 200 patients are needed but the study only includes 190 patients, please justify.

- Please specify if the dentist has been calibrated.

- There is no sample size calculation. If this has been done beforehand, it is important to mention it.

Results :

- Put the number of subjects included in the study.

- Make a clearer sentence "Median age was 65 (52-73) years, 65 (34.2%) were women and 83 (43.7%) and 25 (13.2%) had been diagnosed with diabetes and CAD prior to study recruitment, respectively."

- Add "Male" in table 1

- In table 1 : a multitude of variables were not discuss in the main text.

- How many residual teeth had periapical lesions?

- What were the origins of each bacteremia?

- A radiographic reassessment was made during follow-up?

- Figure 1 does not appear in the text.

- Add table for multivariable analysis

Discussion

- The implementation of individual prophylaxis reduces the risk of bacteremia.

- Did patients receive oral health education prior to follow-up?

Several points should be made clear in this article.

Did the patients have regular follow-up with a dentist?

When did the bacteremia occur? after surgeries? brushing? dental care?

Reviewer #2: Your article is interesting because it evaluates, through a prospective cohort study and with original results, the influence of oral health as risk factors for adverse outcomes in CKD patients transitioning from conservative treatment to maintenance dialysis and transplantation.

The study is well conducted, clear and synthetic with good statistical use. Your study is original because it shows, unlike most other studies, that the level of oral health, only evaluated radiographically (...) is independently associated with mortality and incident MACEs during a followup up of three years in CKD patients transitioning to maintenance dialysis and / or kidney transplantation.

However, it would be advisable to try to explain / justify this first conclusion and in particular by evaluating and exposing the biases of your study. This is one of the main pitfalls of your manuscript.

It would be appropriate to offer a clinically “practical” conclusion and not just to report the conclusions of your statistical dependencies / independence. What is the added value of your study and how it may or may not lead to a change in practices?

Introduction: the rationale and justification of your study is well done. Many studies are cited and report the close link between oral health and chronic kidney disease.

Materials and methods: Your study is a pre-specified report from the CADKID study. Has there been a scientific valuation of this CADKID study to date?

Please specify the duration of patient recruitment

Please specify the power and the expected result for the calculation of the minimum number of individuals to be included

Why did you not study the variability of the reading of your measurements by tests of feasibility and inter and intra-observer variability of the PTI evaluation? (kappa, fisher)

Statistics: How did you statistically manage the 27 patients who were referred to a dentist for full dental assessment? How was ITP defined for these patients?

Specify your confounding factors and how they were eliminated in the causality and independence analysis.

Results: It is surprising to note that you did not find any influence between oral health and diabetes in your study. “There were no significant differences between patients with or without diabetes in any of the DPR measures”. Try to explain this briefly in your discussion.

Why didn't you statistically compare the "dialysis" and "kidney transplant" groups? This could be of interest to the potential importance of risk factors for deterioration of the disease.

Please provide free access to your data stating "The association between PTI and

outcomes also remained significant when the models were adjusted with age and dentist referral following the DPR (data not shown). "

6. PLOS authors have the option to publish the peer review history of their article (what does this mean?). If published, this will include your full peer review and any attached files.

Reviewer #1: No

Reviewer #2: **Yes: **LAN Romain

---

## [Author Response · Author response to Decision Letter 0]

19 Aug 2021

Editor Denis Bourgeois

PlosOne

August 19, 2021

Re: #MS PONE-D-21-11023 Dental Health Assessed Using Panoramic Radiograph and Adverse Events in Chronic Kidney Disease Stage 4-5 Patients Transitioning to Dialysis and Transplantation –A Prospective Cohort Study.

Dear Professor Bourgeois,

Thank you for your letter of August 4 concerning this manuscript. In our prospective observational study we investigated the association between radiographically assessed oral health at baseline and all-cause and cardiovascular mortality, major adverse cardiovascular events and symptomatic bacteremia during a 3-year follow-up in CKD stage 4-5 patients transitioning to dialysis and transplantation. Radiographically assessed dental health was independently associated with all-cause and cardiovascular mortality and MACEs but not with the incidence of bacteremia during follow-up. 

We believe that that we have been able to respond to each of the comments of both reviewers, as outlined below, and are therefore grateful for the opportunity to resubmit the revised manuscript to in PlosOne.

All authors have approved submission of the manuscript, and the manuscript has not been published and is not being considered for publication elsewhere. The authors do not have any conflicts of interest in connection with the submitted article.

Yours sincerely,

Adjunct professor Mikko Järvisalo, MD, PhD

Intensive care unit, Turku University Hospital

Building 18, TG3B 

PO Box 52, 20521 Turku, Finland

Email: mikko.jarvisalo@tyks.fi

Phone: +358-2-3130049

Reviewer #1: Thank you for letting review this interesting article.

Introduction :

- It would be wise to include more recent bibliographic references.

We have made efforts to find more recent bibliographic references and added them to the revised manuscript. The number of references has risen from 18 to 25 and 16/25 (64%) of the references in the revised manuscript have been published between 2012-2020. The text in the Introduction section has been slightly revised due to inclusion of new references (Introduction, page 3, first paragraph, lines 5-10)

- Only periodontal diseases are discussed, whereas bacteremia origins are not limited to periodontal diseases.

We have included a mention on previous studies showing that stage 4-5 CKD patients have a significantly higher risk for hospitalization due to infection of any source including bloodstream infections and carry significantly higher risk for infection related mortality compared to subjects with more preserved kidney function. (Introduction, page 3, second paragraph, lines 18-20)

Methods :

- You state that 200 patients are needed but the study only includes 190 patients, please justify.

This part of the Methods should have been written more accurately and has been revised to avoid misinterpretation.

210 consecutive patients referred to the predialysis outpatient clinic of Turku University Hospital were recruited between August 2013 and September 2017 to the Chronic Arterial Disease, quality of life and mortality in chronic KIDney injury (CADKID) –study. The study population target of the main CADKID study was set to a minimum of 200 patients in the beginning of the recruitment. 

We have revised the Methods section accordingly to clarify this issue (Methods, page 4, first paragraph, lines 3-6 and second paragraph, lines 12-16)

- Please specify if the dentist has been calibrated.

Intra- and inter-observer variabilities were assessed by reanalyzing (after nine months) 40 randomly selected DPR images blinded to the results of the first analysis. We have included data on intra- and inter-observer intraclass correlation coefficients and mean differences between PTI measurements in the revised manuscript. The intra- and inter-observer variabilities were acceptable (page 5, paragraph 3, lines 14-17)

- There is no sample size calculation. If this has been done beforehand, it is important to mention it.

The CADKID study has many sub-studies and examines many primary and secondary end-points and associated risk factors. Therefore, sample size calculations could not be performed in the beginning of the study for all sub-studies such as the present study. We have included this point to the Methods and the limitations section of Discussion (Methods, page 4, second paragraph, lines 12-14 and Discussion, page 11, second paragraph, lines 16-17) 

Results :

- Put the number of subjects included in the study.

Done as requested (Results, page 7, first paragraph, line 2)

- Make a clearer sentence "Median age was 65 (52-73) years, 65 (34.2%) were women and 83 (43.7%) and 25 (13.2%) had been diagnosed with diabetes and CAD prior to study recruitment, respectively."

We have revised the sentence (Results, page 7, first paragraph, lines 4-6)

- Add "Male" in table 1

We have included the number of male subjects in Table 1

- In table 1 : a multitude of variables were not discuss in the main text.

We have included several of the variables given in Table 1 to the main text (Results). However, we have aimed to give data on the most important variables in the text, whereas, more comprehensive data are shown in the Table 1. (Results, page 7, first paragraph, lines 4-8)

- How many residual teeth had periapical lesions?

Of the total 109 residual roots detected in the current study 50% (55) were interpreted as having concomitant periapical periodontitis. We have included this point to the revised manuscript (page 7, paragraph 1, lines 10-11).

- What were the origins of each bacteremia?

We have manually re-reviewed all the individual patients records in the patients with blood culture positive bacteremia during follow-up. The number of patients with incident bacteremias was reduced to 22 and has been corrected to the abstract and results as have been all related results. Only two of the bacteremias were considered to indisputably be of oral origin. We have included the sources of the first incident bacteremia to the revised manuscript. The source of infection was skin/soft tissues in 10 (45%), urinary tract in 3 (14%), abdominal in 5 (23%), catheter-related bloodstream infection 1 (5%), dental 2 (9%) and respiratory in 1 (5%) patients, respectively. (Results, page 8, first paragraph, lines 1-5).

- A radiographic reassessment was made during follow-up?

No radiographic re-assessments were unfortunately included in the study protocol. We have included a mention on this in the limitations section of the revised manuscript (Discussion, page 11, second paragraph, lines 4-5).

- Figure 1 does not appear in the text.

Thank you. It was missing due to an unfortunate error. We have included a reference to Figure 1 (Results, page 7, first paragraph, lines 9-10)

- Add table for multivariable analysis

We have included a table describing the univariate and multivariable results (Table 3). Furthermore, we have also simplified the multivariable models and revised the statistical methods and results accordingly after consulting an experienced biostatistician. (Methods, page 6, second paragraph, lines 24-25 and page 6, first paragraph, lines 1-8; Results, page 8, second paragraph, lines 4-13; Acknowledgements; Table 3)

Discussion

- The implementation of individual prophylaxis reduces the risk of bacteremia.

Five of the patients were on continuous prophylactic antibiotics (Trimethoprim, recurrent urinary tract infection, n=1; sulfatrimethoprim, Pneumocystic jirovecii prophylaxis, n=1; phenoxymethylpenicillin n=1, recurrent cellulitis; ciprofloxacin, recurrent urinary tract infection, n=1; Clindamycin, recurrent cellulitis, n=1). Unfortunately, we have no data on occasional antibiotics administered such as one-time prophylaxis associated with operations or procedures. We have included these data to the revised manuscript and shortly discussed this issue. One of the patients on continuous prophylaxis had an incident bacteremia during follow-up. (Results, page 8, first paragraph, lines 7-11; Discussion, page 10, paragraph 4, line 26 – page 11, paragraph 1, lines 1-3)

- Did patients receive oral health education prior to follow-up?

All patients receive verbal oral health education/are interviewed on potential dental problems at our kidney centre predialysis out patient clinic and are recommended to visit a dentist yearly. After the results of the present study we are, however, changing our routines concerning CKD stage 4-5 patients which will all to be referred to DPR and proper dental examination at our hospital starting 2022 after their first visit to our predialysis outpatient clinic. (Methods, page 4, third paragraph, lines 18-21 and Discussion, page 11, paragraph 2, lines 6-8) 

Several points should be made clear in this article.

Did the patients have regular follow-up with a dentist?

All patients at our kidney centre predialysis outpatient clinic have been advised to visit a dentist yearly. At the time of the study most patients visited a private sector dentist and only those with obvious dental symptoms/imaging findings were referred to a dentist at the research hospital. No regular follow-up by a dentist was included in the study protocol and DPR was only systematically assessed at study baseline. No data was available on the visits of the patients to private sector dentists. (Methods, page 4, third paragraph, lines 18-21 and and Discussion, page 11, paragraph 2, lines 6-10) 

After the results of the present study, we are changing our routines concerning CKD stage 4-5 patients which will all to be referred to dental examination at our hospital starting 2021-2022.

When did the bacteremia occur? after surgeries? brushing? dental care?

We have included the time to bacteremia [796 (IQR 432-883, range 31-1056) days] in the manuscript. Unfortunately, we have no exact data on the situations when bacteremia emerged. However, only a minor proportion of the bacteria found were considered potentially to be of dental origin and to our knowledge none emerged acutely/sub-acutely after dental care. We have included data on sources of infection in the patients with incident blood culture positive sepsis during follow-up. (Results, page 8, first paragraph, lines 1-6) 

Reviewer #2: Your article is interesting because it evaluates, through a prospective cohort study and with original results, the influence of oral health as risk factors for adverse outcomes in CKD patients transitioning from conservative treatment to maintenance dialysis and transplantation.

The study is well conducted, clear and synthetic with good statistical use. Your study is original because it shows, unlike most other studies, that the level of oral health, only evaluated radiographically (...) is independently associated with mortality and incident MACEs during a followup up of three years in CKD patients transitioning to maintenance dialysis and / or kidney transplantation.

However, it would be advisable to try to explain / justify this first conclusion and in particular by evaluating and exposing the biases of your study. This is one of the main pitfalls of your manuscript.

We have revised the limitations section of discussion in our revised manuscript for better evaluation of potential biases and merits of our data.

The main limitation of our study was that DPR imaging was only performed in the beginning of the study and is therefore a cross-sectional measure, in spite of the prospective patient follow-up. Furthermore, the study protocol did not include a clinical dental assessment by a dentist. All patients receive verbal oral health education/are interviewed on potential dental problems at the predialysis outpatient clinic and are recommended to visit a dentist yearly. At the time of the study most patients were instructed to visit a private sector dentist and only those with obvious dental symptoms/imaging findings were referred to a dentist at the research hospital. No data was available on the visits of the patients to private sector dentists. The number of patients included in the analysis is somewhat limited, but the CADKID study recruited all consecutive willing CKD stage 4-5 patients referred to the Turku University Hospital Kidney Centre between 2013 and 2017 and 90% of the CADKID study patients had DPR data. The patients were followed-up prospectively and regularly at our Kidney Center, and the data are of high quality and comprehensive. Due to the limited number of adverse events during follow-up variables included in the multivariable models had to be limited to avoid overfitting which may increase the risk for unrecognized confounders. As power calculations were not performed, it is possible that the study was underpowered for some of the analyses performed. The single center observational study design does not allow determination of causality and increases the potential for selection bias. The findings, however, were distinct for all outcome measures and limited sample size is not likely to detract the validity of our main findings concerning the association between PTI and adverse outcomes. (Discussion, page 11, second paragraph, lines 4-20)

It would be appropriate to offer a clinically “practical” conclusion and not just to report the conclusions of your statistical dependencies / independence. What is the added value of your study and how it may or may not lead to a change in practices?

We have included a conclusion that reflects the clinical implications of our current findings. 

Our current data suggest that DPR imaging offers a noninvasive and inexpensive tool for dental health associated mortality and cardiac risk stratification in patients with advanced CKD and should be included in the routine clinical assessment of CKD stage 4-5 patients. (Discussion, page 12, first paragraph, lines 4-6) 

Introduction: the rationale and justification of your study is well done. Many studies are cited and report the close link between oral health and chronic kidney disease.

We thank the reviewer for this kind comment.

Materials and methods: Your study is a pre-specified report from the CADKID study. Has there been a scientific valuation of this CADKID study to date?

The CADKID study is registered at ClinicalTrials.org. Several manuscripts from the CADKID study data have been published to date in indexed peer-reviewed journals: 

Lankinen R et al. Am J Nephrol. 2020;51(9):726-735. doi: 10.1159/000509582.

Hellman T et al. BMC Cardiovasc Disord. 2020;20(1):437. doi: 10.1186/s12872-020-01719-3.

Lankinen R et al. BMC Nephrol 2021;22(1):50. doi: 10.1186/s12882-021-02251-y.

Hakamäki M et al. Nephron. 2021;145(1):71-77. doi: 10.1159/000511451.

Hakamäki M et al. Blood Purif. 2021;50(3):347-354. doi: 10.1159/000510984

Hellman T et al. Diabetes Res Clin Pract. 2021;171:108559. doi: 10.1016/j.diabres.2020.108559.

Please specify the duration of patient recruitment

The duration of the CADKID study patient recruitment is described in the methods section of the revised manuscript.

210 consecutive patients referred to the predialysis outpatient clinic of Turku University Hospital were recruited between August 2013 and September 2017 to the Chronic Arterial Disease, quality of life and mortality in chronic KIDney injury (CADKID) –study. (Methods, page 4, first paragraph, lines 3-11)

Please specify the power and the expected result for the calculation of the minimum number of individuals to be included

The CADKID study has many sub-studies and many primary and secondary end-points. Therefore, sample size calculations could not be performed for all sub-studies such as the present study. We have included this point to the Methods and the limitations section of Discussion (Methods, page 4, second paragraph, lines 12-14 and Discussion, page 11, second paragraph, lines 16-17) 

Why did you not study the variability of the reading of your measurements by tests of feasibility and inter and intra-observer variability of the PTI evaluation? (kappa, fisher)

Intra- and inter-observer variabilities were assessed by reanalyzing after nine months 40 randomly selected DPR images blinded to the results of the first analysis. We have included data on intra- and inter-observer intraclass correlation coefficients of PTI measurements in the revised manuscript. The intra- and inter-observer variabilities were acceptable (Methods, page 5, paragraph 3, lines 14-17).

Statistics: How did you statistically manage the 27 patients who were referred to a dentist for full dental assessment? How was ITP defined for these patients?

Patients with dentist referral after DPR imaging were treated statistically similarly as other patients. The aim of the study was to assess whether a radiographic assessment of dental health (PTI) is associated with incident adverse effects. The clinical dental examinations were not conducted systematically as a part of the study protocol and therefore no data on this dental assessment are available.

PTI was defined similarly as in other patients. However, we found it important to adjust for the dentist referral when studying the association between PTI and adverse events as dental referral (and potential treatment) was considered to potentially be a confounding factor. Patients that were referred to the dentist following DPR had significantly higher PTI compared to others [1 (1-3) vs. 1 (0-2) ,p=0.003]. We have included these data to the revised manuscript (Methods, page 6, second paragraph, lines 12-13; Results, page 7, second paragraph, lines 25-26, Results page 8, second paragraph, lines 24-27)

Specify your confounding factors and how they were eliminated in the causality and independence analysis.

A previous analysis of the CADKID study data has shown that age, diabetes and coronary artery disease are associated with mortality in the cohort (Lankinen R et al. Am J Nephrol. 2020;51(9):726-735. doi: 10.1159/000509582) and were therefore included as covariates in the respective multivariable Cox regression models assessing the independent association between PTI and mortality.

We have included a table on the univariate and multivariable models. The variables included as covariates in the respective models were, Model 1: age, diabetes, coronary artery disease, PTI; Model 2: age, diabetes, coronary artery disease, PTI, treatment modality for CKD at the end of study or death. 

With so few events in the cohort, adjustment for multiple covariates in a single multivariable model could not be done as including these variables in the models (with very few events) would have led to increased risk of overfitting and the statistical models would not have been sound. Therefore, the potential effect of dentist referral on the association between PTI mortality and MACES was examined in respective models with age as a covariate. (Methods, page 6, second paragraph, lines 4-13; Discussion, page 11, second paragraph, lines 14-16; Table 3)

Results: It is surprising to note that you did not find any influence between oral health and diabetes in your study. “There were no significant differences between patients with or without diabetes in any of the DPR measures”. Try to explain this briefly in your discussion.

The reason for this finding is unclear. It is possible that dental health is managed more properly in diabetic patients who also have regular outpatient controls as diabetes is considered a risk factor for oral health. We have included a mention on the issue in the revised manuscript (Discussion, page 10, third paragraph, lines 13-16) 

Why didn't you statistically compare the "dialysis" and "kidney transplant" groups? This could be of interest to the potential importance of risk factors for deterioration of the disease.

The primary aim of the study was to examine whether DPR findings are associated with incident death and MACEs in advanced CKD. The prognosis of kidney transplant patients is known to exceed the prognosis of patients on maintenance dialysis. Those that are accepted to the transplantation waiting list are usually younger and have less comorbidities compared to other maintenance dialysis patients. In the present cohort, none of the patients that received a transplant during follow-up deceased whereas 24 (25%) of those who were on maintenance dialysis until end of follow-up (death or 36 months) died. There were, however, no significant differences in PTI assessed at study baseline between patients on different treatment modalities for CKD at the end of follow-up [conservative treatment: 1 (0-3), peritoneal dialysis: 1 (0-2), hemodialysis: 1 (0-2), transplantation: 1 (0-2), p=0.49]. We have included these data to the revised manuscript (Results, page 7, second paragraph, lines 18-21). PTI was independently associated with mortality and MACEs after controlling for treatment group in the multivariable model. 

Please provide free access to your data stating "The association between PTI and outcomes also remained significant when the models were adjusted with age and dentist referral following the DPR (data not shown). "

We have included these data in the results section (Results, page 8, third paragraph, lines 24-27)

We would like to thank the reviewers for constructive criticism and comments and feel that the manuscript has improved by the changes made. Therefore, we hope that you will consider this revised manuscript acceptable for publication in PlosOne.

---

## [Decision Letter · Decision Letter 1]

13 Sep 2021

**Comments to the Author**

1. If the authors have adequately addressed your comments raised in a previous round of review and you feel that this manuscript is now acceptable for publication, you may indicate that here to bypass the “Comments to the Author” section, enter your conflict of interest statement in the “Confidential to Editor” section, and submit your "Accept" recommendation.

Reviewer #1: All comments have been addressed

Reviewer #2: All comments have been addressed

2. Is the manuscript technically sound, and do the data support the conclusions?

Reviewer #1: Yes

Reviewer #2: Partly

3. Has the statistical analysis been performed appropriately and rigorously? 

Reviewer #1: Yes

Reviewer #2: Yes

4. Have the authors made all data underlying the findings in their manuscript fully available?

Reviewer #1: Yes

Reviewer #2: Yes

5. Is the manuscript presented in an intelligible fashion and written in standard English?

Reviewer #1: Yes

Reviewer #2: Yes

6. Review Comments to the Author

Reviewer #1: Thank you for the changes made in a rigorous manner. Limitations of the study are well highlighted which is important. Conclusion provides recommendations for improving patient management

Reviewer #2: Thank you for this new submission which made important changes in line with our comments. Your document is much clearer and more precise (introduction - statistics - results).

Methods - study protocol: there were “no regular follow-up by a dentist in the study protocol”.

Specify that this is a bias in your discussion, especially on the data identified in the follow-up. Especially since you don't take new x-rays.

Methods - dental radiographs: Reading your paragraph, I understand that in the end only 171 patients were able to benefit from a PRD? So specify the final inclusion number (190? 171?).

Specify the statistical test used for the inter-intra observer variance.

Discussion: Provide a hypothesis / justification / consequence to your new result regarding the outcome of bacteremia, and the weak relation to oral origin. This is major information.

There is, for me, a difference between individual dental prophylaxis and antibiotic prophylaxis as you are only discussing. To review. Because the bacteremia has a clear influence depending on the availability of individual dental prophylaxis or not (many publications on this subject).

"No significant differences between patients with or without diabetes in any of the DPR measures": I think the rationale is rather that poor periodontal health is a risk factor for worsening diabetes, and therefore that regular information and monitoring are needed in these patients, and not the other way around as you assume (although diabetes is also a risk factor for periodontal disease).

7. PLOS authors have the option to publish the peer review history of their article (what does this mean?). If published, this will include your full peer review and any attached files.

Reviewer #1: **Yes: **Camille INQUIMBERT

Reviewer #2: **Yes: **LAN Romain

---

## [Author Response · Author response to Decision Letter 1]

15 Sep 2021

Editor Denis Bourgeois

PlosOne

September 14, 2021

Re: #MS PONE-D-21-11023 Dental Health Assessed Using Panoramic Radiograph and Adverse Events in Chronic Kidney Disease Stage 4-5 Patients Transitioning to Dialysis and Transplantation –A Prospective Cohort Study.

Dear Professor Bourgeois,

Thank you for your letter of September 14 concerning this manuscript. In our prospective observational study we investigated the association between radiographically assessed oral health at baseline and all-cause and cardiovascular mortality, major adverse cardiovascular events and symptomatic bacteremia during a 3-year follow-up in CKD stage 4-5 patients transitioning to dialysis and transplantation. Radiographically assessed dental health was independently associated with all-cause and cardiovascular mortality and MACEs but not with the incidence of bacteremia during follow-up. 

We believe that that we have been able to respond to each of the comments of the reviewers, as outlined below, and are therefore grateful for the opportunity to resubmit the revised manuscript to PlosOne.

All authors have approved submission of the manuscript, and the manuscript has not been published and is not being considered for publication elsewhere. The authors do not have any conflicts of interest in connection with the submitted article.

Yours sincerely,

Adjunct professor Mikko Järvisalo, MD, PhD

Intensive care unit, Turku University Hospital

Building 18, TG3B 

PO Box 52, 20521 Turku, Finland

Email: mikko.jarvisalo@tyks.fi

Phone: +358-2-3130049

Reviewer #1: Thank you for the changes made in a rigorous manner. Limitations of the study are well highlighted which is important. Conclusion provides recommendations for improving patient management

Reviewer #2: Thank you for this new submission which made important changes in line with our comments. Your document is much clearer and more precise (introduction - statistics - results).

Methods - study protocol: there were “no regular follow-up by a dentist in the study protocol”.

Specify that this is a bias in your discussion, especially on the data identified in the follow-up. Especially since you don't take new x-rays.

We agree with the reviewer and have revised the Limitations section of the discussion accordingly to include this point. (Page 11, Paragraph 2, Lines 254-258)

Methods - dental radiographs: Reading your paragraph, I understand that in the end only 171 patients were able to benefit from a PRD? So specify the final inclusion number (190? 171?).

The correct number is 190. We have revised the text to clarify the number of patients with DPR and included in the analyses. (Page 5, Paragraph 2, Lines 99-104) 

Specify the statistical test used for the inter-intra observer variance.

We have included this point to the Statistical methods section of the manuscript (Page 6, Paragraph 3, Lines 133-136)

Discussion: Provide a hypothesis / justification / consequence to your new result regarding the outcome of bacteremia, and the weak relation to oral origin. This is major information.

We have included the point that only a minor proportion of the bacteremias were of oral origin, which also makes it understandable that we did not observe an association between DPR findings and incident bacteremia. (Page 10, Paragraph 4, Lines 242-244) 

There is, for me, a difference between individual dental prophylaxis and antibiotic prophylaxis as you are only discussing. To review. Because the bacteremia has a clear influence depending on the availability of individual dental prophylaxis or not (many publications on this subject).

We agree with the reviewer but unfortunately these data were not available for the current study. We have included this point to the revised manuscript. (Page 11, Paragraph 1, Lines 250-253)

"No significant differences between patients with or without diabetes in any of the DPR measures": I think the rationale is rather that poor periodontal health is a risk factor for worsening diabetes, and therefore that regular information and monitoring are needed in these patients, and not the other way around as you assume (although diabetes is also a risk factor for periodontal disease).

The reviewer is right. We have revised the text accordingly. (Page 10, Paragraph 3, Lines 233-236) 

We would like to thank the reviewers for constructive criticism and comments and feel that the manuscript has improved by the changes made. Therefore, we hope that you will consider this revised manuscript acceptable for publication in PlosOne.

---

## [Editor Report · Decision Letter 2]

17 Sep 2021

Dental Health Assessed Using Panoramic Radiograph and Adverse Events in Chronic Kidney Disease Stage 4-5 Patients Transitioning to Dialysis and Transplantation –A Prospective Cohort Study.

PONE-D-21-11023R2

Dear Dr. Järvisalo,

We’re pleased to inform you that your manuscript has been judged scientifically suitable for publication and will be formally accepted for publication once it meets all outstanding technical requirements.

Kind regards,

Denis Bourgeois

Academic Editor

PLOS ONE
---

## [Editor Report · Acceptance letter]

22 Sep 2021

PONE-D-21-11023R2 

Dental Health Assessed Using Panoramic Radiograph and Adverse Events in Chronic Kidney Disease Stage 4-5 Patients Transitioning to Dialysis and Transplantation –A Prospective Cohort Study. 

Dear Dr. Järvisalo:

I'm pleased to inform you that your manuscript has been deemed suitable for publication in PLOS ONE. Congratulations! Your manuscript is now with our production department. 

Kind regards, 

on behalf of

Professor Denis Bourgeois 

Academic Editor

PLOS ONE